# Multilayer Graphene Oxide Supported ZIF-8 for Efficient Removal of Copper Ions

**DOI:** 10.3390/nano12183162

**Published:** 2022-09-13

**Authors:** Xifeng Lv, Yishi Zhang, Xiaodong Wang, Libing Hu, Chunhui Shi

**Affiliations:** 1College of Chemistry and Chemical Engineering, Tarim University, Alar 843300, China; 2Engineering Laboratory of Chemical Resources Utilization in South Xinjiang of Xinjiang Production and Construction Corps, Alar 843300, China; 3State Key Laboratory of Organic-Inorganic Composites, Beijing University of Chemical Technology, Beijing 100029, China

**Keywords:** multilayer graphene oxide, ZIF-8, in situ growth, heavy metal adsorption, adsorption kinetics

## Abstract

**Highlights:**

**Abstract:**

To address the performance deterioration of ZIF-8 for the adsorption of copper ions caused by powder volume pressure and particle aggregation, we employed multilayer graphene oxide (MGO) as a support to prepare composite adsorbents (MGO@ZIF-8) by using the in situ growth of ZIF-8 on MGO. Due to a good interfacial compatibility and affinity between ZIF-8 and graphene nanosheets, the MGO@ZIF-8 was successfully prepared. The optimal Cu^2+^ adsorption conditions of MGO@ZIF-8 were obtained through single factor experiments and orthogonal experiments. Surprisingly, the Cu^2+^ adsorption capacity was significantly improved by the integration of MGO and ZIF-8, and the maximum Cu^2+^ adsorption capacity of MGO@ZIF-8 reached 431.63 mg/g under the optimal adsorption conditions. Furthermore, the kinetic fitting and isotherm curve fitting confirmed that the adsorption law of Cu^2+^ by MGO@ZIF-8 was the pseudo-second-order kinetic model and the Langmuir isotherm model, which indicated that the process of Cu^2+^ adsorption was monolayer chemisorption. This work provides a new approach for designing and constructing ZIF-8 composites, and also offers an efficient means for the removal of heavy metals.

## 1. Introduction

In recent years, the development of modern industry has led to the increasingly serious pollution of heavy metal ions (such as chromium, lead, and copper), which has caused serious harm to human health and social development [1,2]. Some studies have noted that when the intake of copper exceeds standard levels, it will generate a burden to human organs, causing metabolic disorders and liver cirrhosis [3,4]. Therefore, how to effectively reduce and eliminate copper ion (Cu^2+^) pollution has become an important social and economic issue [5,6].

Currently, chemical deposition, ion exchange, adsorption and membrane separation have been used to dislodge Cu^2+^ from wastewater [7,8]. Among them, adsorption has been regarded as one of the most efficient strategies, since it has advantages of high-removal efficiency, ease of operation, and practical, low-budget and low environmental impact [9,10]. In particular, the elimination of Cu^2+^ adsorption process can provide additional technical and economic benefits. Therefore, it is necessary to study the application of novel adsorbents to consolidate their industrial application [11,12,13]. In this direction, it has been reported that two-dimensional (2D) nanomaterials show potential as promising adsorbents for heavy metal removal [14,15,16]. For instance, 2D graphene oxide (GO) with large surface area includes reactive functional groups (carboxyl, hydroxyl, and carbonyl groups) and exhibits high affinity with metal ions [17]. However, the adsorption capacity of GO is mainly restricted to the number of surface functional groups, thus there is a high need to improve the Cu^2+^adsorption of MGO sheets [18,19].

In addition, metal–organic frameworks (MOFs) have been widely used as a hazardous remediation agent in heavy metal precipitation and adsorption because of their distinct porosity, adsorption capacity, high specific surface areas, tunable porous structure, and modifiability [20]. However, the analysis and discussion of the interaction between adsorbent and adsorbed material, as well as the adsorption mechanism, are still insufficient [21,22,23]. It should be noted that the structure of the adsorbent (e.g., molecular particle size, functional groups, molecular polarity and solubility) has an effect on adsorption performance [24,25]. Due to their crystalline character, MOFs accumulate in powder form. Investigations into MOFs have shown that their ability to remove Cu^2+^ is largely dependent on the duality of their particles [26,27,28,29]. However, MOF powders tend to accumulate, which greatly limits the exposure of adsorption sites and leads to a decrease in adsorption capacity. Inspired by this, the conjugation of MOFs with suitable substrates is considered as a feasible approach to meet the needs of the application [30,31,32]. Herein, we propose a facile strategy for the in situ growth of ZIF-8 between laminated graphene channels to form ZIF-8/graphene composites, whereby the MGO interlayer supports and disperses the attachment of ZIF-8 particles, and is well retained. The results show that the porous property of graphene provides channels for water transport, and that MGO@ZIF-8 can be used to adsorb Cu^2+^ efficiently. This work offers a new adsorbent for the removal of heavy metals and can broaden its application for use in environmental treatment.

## 2. Experimental

### 2.1. Materials

MGO was purchased from Carbon Feng Technology of Suzhou, methyl alcohol was bought from Tianjin Yongda Chemical Reagent Co., Ltd. (Tianjin, China), 2-Methylimidazole (2-Melm) was obtained from Tokyo Kasei Kogyo Co. of Japan, and sodium dodecylbenzene sulfonate (DBS) was gained from J&K Technology Co. of China in Xiamen city. Polyvinyl pyrrolidone (PVP) was purchased from Tianjin Feng Chuan Chemical Reagent Technology Co., Ltd. (Tianjin, China). All materials employed in this study were of analytical grade without further purification. Ultra-pure water was used in the experiments. A magnetic constant temperature stirrer, MYP19-2 type, was obtained from Shanghai Meiyingpu Instrument Manufacturing Co., Ltd. (Shanghai, China).

### 2.2. Synthesis of MGO@ZIF-8

As shown in Figure 1, DBS was ultrasonically dissolved in 120 mL methyl alcohol, and 60 mg of MGO was added after dissolution to prepare the MGO suspension. Then, 4 mg of PVP was poured into the suspension, and after subjecting to ultrasound for 1 h, the supernatant was removed and centrifuged for 10 min. After that, the precipitation obtained was redispersed into methyl alcohol, and 5.622 g of 2-Melm was added into the dispersion solution until it was completely dissolved.. Subsequently, the dispersion was placed on a magnetic stirrer at moderate speed, and methyl alcohol of zinc nitrate (2.572 g) was slowly added at 298 K and stirred for 1 h. Finally, the precipitation was collected by centrifugation, and the activated MGO@ZIF-8 was prepared after vacuum drying.

### 2.3. Assessment of MGO@ZIF-8 for Cu^2+^ Adsorbing Capacity

In the process of undertaking the adsorption experiment, the effects of different pH, adsorption time, adsorption temperature and the amount of adsorbent were investigated for MGO@ZIF-8, and the adsorption of Cu^2+^ was observed. An appropriate amount of copper sulfate was dissolved in deionized water to prepare a certain concentration of copper ion solution, and we then adjusted the effect of different pH (1–7) on the adsorption amount with hydrochloric acid and sodium hydroxide at 35 °C. The effect of different adsorption times (0–110 min) on the adsorption capacity was investigated every 10 min at pH = 6 at 35 °C. The effect of different temperatures (25–50 °C) on the adsorption capacity was investigated at pH = 6. The effect of different adsorbent dosages (1–8 mg) on the adsorption capacity was investigated at pH = 6 and 35 °C. The corresponding adsorption capacity was obtained according to adsorption capacity shown in Formula (1) [33,34,35].
(1)$$q_{t}=\frac{\left(c_{0}-c_{t}\right)}{M}V$$
where *c*_0_ (mg/L) is the solution initial concentration; *q_t_* (mg/g) is adsorption quantity; *c_t_* (mg/L) is the residual concentration of the target substance at time T; *M* (mg) is the mass of adsorbent; and *V* (L) is the volume of the solution.

### 2.4. Standard Curve of Cu^2+^

Cu^2+^ solutions with different concentration gradients were prepared, and the concentration of copper ions was detected according to a flame atomic absorption instrument. The standard curve of Cu^2+^ was drawn by software, as shown in Figure 2.

### 2.5. Adsorption Kinetics

In the study, the dynamics were fitted Lagergren quasi-first-order kinetics equation and quasi-second-order kinetics equation [36,37,38], as shown below.

Lagergren quasi-first order kinetic equation:(2)$$\ln(Q_{e}-Q_{t})=\ln Q_{e}-K_{1}t$$

Lagergren quasi-second-order kinetics equation:(3)$$\frac{t}{Q_{t}}=\frac{1}{Q_{e}{}^{2}K_{2}}+\frac{t}{Q_{e}}$$
wherein *Q_e_* (mg/g) is the adsorption capacity at equilibrium; *Q_t_* (mg/g) expresses the adsorption capacity at any moment; *K*_1_ (min^−1^) represents the constant of Lagergren first-order kinetics; and *K*_2_ (g/(min·mg)) denotes the constant of the Lagergren second-order kinetics.

### 2.6. Adsorption Isotherms

The adsorption isotherms were employed based on the Freundlich adsorption isotherm model and Langmuir adsorption isotherm model [39,40,41]. The Langmuir adsorption isotherm is shown as:(4)$$\frac{C_{e}}{q_{e}}=\frac{1}{q_{max}k_{L}}+\frac{C_{e}}{q_{ma}{}_{x}}$$

In Equation (4), *q_e_* (mg/g) is named as amount of metal ions adsorbed by adsorbent per unit weight; *q_max_* (mg/g) represents complete single-layer coverage of the corresponding maximum adsorption capacity; *C_e_* (mg/L) denotes the equilibrium concentration of metal ions; and *K_L_* (L/mg) means a constant for the free energy of adsorption. The slope and intercept of *C_e_/q_e_* versus *c_e_* is able to derive the values of *q_max_* and *k_L_*, respectively.

The Freundlich adsorption isotherm is shown as:(5)$$q_{e}=k_{F}C_{e}^{1/n}$$

In Equation (5), *q_e_* (mg/g) indicates the equilibrium adsorption capacity, and *K_F_* and *n* express the respective Freundlich constants for adsorption capacity and adsorption intensity. The adsorption process is favorable if *1/n* is smaller than unity. The slope and intercept of log *q_e_* versus log *C_e_* can calculate the values of *K_F_* and *n*, respectively.

## 3. Results and Discussion

### 3.1. Characterization of Adsorbents

#### 3.1.1. SEM and AFM Analysis

The SEM and AFM images of the samples are displayed in Figure 3. As shown in Figure 3a, the MGO was a superposition of multi-layered structures. ZIF-8 grew uniformly between the graphene layers with a crystal size distribution between 50 and 150 nm (Figure 3b). Graphene provided a supporting framework for ZIF-8 [42,43], preventing the reduction in the specific surface area caused by the volume pressure of the ZIF-8 powder. This porous structure can increase the active sites for metal ion adsorption and improve adsorption performance. It can be seen from Figure 3b that ZIF-8 was uniformly distributed on the hierarchical surface of the MGO, and the fully exposed surface could provide more attachment sites for copper ions, which effectively improves the adsorption performance of the composites [44,45].

#### 3.1.2. XRD Diffraction and FT-IR Spectroscopy

In order to verify the phases and components of the samples, they were characterized by XRD (Figure 4a). As can clearly be seen, the XRD patterns of ZIF-8 showed that strong diffraction peaks at 7.13°, 10.19°, 7.05° and 17.84°, corresponded to (001), (002), (112) and (222) crystal planes of ZIF-8, suggesting that the pure ZIF-8 was prepared successfully. Moreover, a diffraction peak in MGO located at 2θ value of 26.21°, which corresponded well to the (111) crystal plane of MGO, and importantly, no other diffraction peaks of other crystal phases were observed [46]. Amazingly, the characteristic peaks of MGO and ZIF-8 simultaneously appeared in the acquired XRD patterns of MGO@ZIF-8, suggesting that the desirable MGO@ZIF-8 composite was successfully prepared without forming any other new crystals.

The chemical groups of the three samples were further examined according to FT-IR spectroscopy, as shown in Figure 4b. Therein, no peaks appeared at the MGO, however, obvious characteristic peaks at (a) 3431 cm^−1^, (b) 2927 cm^−1^ and (g) 421 cm^−1^ due to N-H stretching, C-H bending vibrations and Zn-N stretching were observed in ZIF-8, respectively. Different from the spectra of the MGO and ZIF-8, the bonding interaction diagram shows that ZIF-8 in MGO@ZIF-8 appeared a “blue shift” phenomenon at the position of (d) 1586 cm^−1^, and moved to (c) 1649 cm^−1^, due to the change of the dipole moment of the C=C and C=N groups after MGO and ZIF-8 were recombined [47].

#### 3.1.3. Specific Surface Area and BET Analysis

Nitrogen adsorption–desorption analysis was used to observe the porous structures of MGO@ZIF-8 based on the Brunauer–Emmett–Teller (BET) method. According to the definition given by the International Society for Theoretical and Applied Chemistry (IUPAC), the nitrogen isotherm of MGO@ZIF-8 belongs to a type IV isotherm (obvious increasing uptake at high P/P_0_), which indicates that the adsorption of MGO@ZIF-8 was as monolayers, as shown in Figure 5, and the BET specific surface area of MGO@ZIF-8 calculated from nitrogen isotherm was found to be 249.7 m^2^/g. In addition, as shown in Figure 6, the central aperture of MGO@ZIF-8 was about 3 nm, and the pore diameter was present at 2–15 nm, confirming that the composite material was mesoporous and had a large specific surface area [48,49]. Therefore, it was also verified that the adsorption performance of MGO@ZIF-8 could be improved by the composite of ZIF-8 and MGO.

#### 3.1.4. XPS Analysis

X-ray photoelectron spectroscopy (XPS) spectra was conducted to further probe the differences in the chemical composition of the composite surface after surface modification, and the interactions between the MGO@ZIF-8 and Cu^2+^. The wide scanning XPS spectra of MGO@ZIF-8 before and after Cu^2+^ adsorption are exhibited in Figure 7a,e, respectively. Compared with the pristine MGO@ZIF-8, it can clearly be seen that the Cu 2p_3/2_ peak presented in the composite of Cu-MGO@ZIF-8, confirming that MGO@ZIF-8 was able to adsorb Cu^2+^. Additionally, the energy spectrum of Zn 2p with absorption peaks at 1045.08 eV and 1022.08 eV fully indicated the existence of Zn 2p_1/2_ and Zn 2p_3/2_ (Figure 7b), suggesting that MGO@ZIF-8 was successfully synthesized [50]. The difference of 23 eV between the two peaks effectively confirmed that the valence state of Zn in MGO@ZIF-8 was 2s, which was consistent with reports in the known relevant literature. Figure 7c displayed the C 1s energy spectrum of MGO@ZIF-8 material at 285.57 eV and 284.85 eV, corresponding to C-N and C=N in dimethylimidazole and organic ligands in ZIF-8 material, respectively. The energy spectrum of O 1s was present in MGO@ZIF-8 (Figure 7d), simultaneously O=C-O at 532.78 eV and Zn-O at 529.86 eV appeared in MGO@ZIF-8, which also indicated that ZIF-8 successfully combined into the layers of MGO. In particular, the energy spectrum of Cu 2p, which was Cu 2p_1/2_ and Cu 2p_3/2_ at 168 eV and 368 eV, respectively, could be explained by the composite material having an adsorption effect on Cu^2+^, and the adsorption process was chemical adsorption.

### 3.2. Adsorption Mechanism

#### 3.2.1. pH Effect on Cu^2+^ Adsorption by MGO@ZIF-8

As is known to all, the pH value of a solution is an important determinant that affects the adsorption of metal ions by adsorbents, as it not only affects the kinds of metals in the solution, but also has a great influence on the surface properties of the adsorbent. Although it has been proved that most MGO adsorbents have good adsorption capacity for Cu^2+^, they depend on pH value, which is not conducive to practical application [51]. In this study, the pH effect on the MGO@ZIF-8 adsorption capacities for Cu^2+^ was researched at pH from 1 to 7 and the results are displayed in Figure 8. Evidently, MGO@ZIF-8 showed different Cu^2+^ adsorption capacity with the change in pH, indicating that wide adaptability was dependent on pH. It can clearly be seen that the Cu2+ adsorption capacity increased with pH increasing from 1 to 6, as the increase of pH in the solution resulted in attenuating the electrical repulsion force, which increased the adsorb possibility of metal ions on the surface of MGO@ZIF-8. When pH = 6, MGO@ZIF-8 delivered the highest adsorption capacity with a Cu^2+^ adsorption mass of 450 mg/g. However, when the pH increased to 7, the Cu^2+^ adsorption capacity of MGO@ZIF-8 decreased, which was mainly caused by the protonation of functional groups in the composite, leading to a positive charge on the material surface, and a competitive combination of hydrogen ions and copper ions occurred in the solution.

#### 3.2.2. Temperature Effect on Cu^2+^ Adsorption by MGO@ZIF-8

For the sake of examining the influence of the temperature effect on Cu^2+^ adsorption, different reaction temperatures were used to study the concentration of Cu^2+^ solution with an adsorption time of 2 h. The study found that the temperature of Cu^2+^ solution for MGO@ZIF-8 adsorption capacity engendered a positive effect, as shown in Figure 9. Under the extension of 25 °C to 35 °C with the same initial concentration of the Cu^2+^ solution, it was obvious that *q*_e_ increased when the temperature was raised. Therefore, the adsorption reaction was conjectured to be endothermic. Raised temperatures expedited the thermal motion of Cu^2+^, creating an increased physical adsorption rate of Cu^2+^ on the MGO@ZIF-8 surface. However, the elevated temperature increased energy that was relied upon within the adsorption process; stable chemical bonds formed between MGO@ZIF-8 and Cu^2+^, working to enhance the *q_e_*. However, the adsorption rate slowed down with the increase in temperature, because the adsorption of the adsorbent material for the solution system was close to equilibrium when the adsorption capacity reached 430 mg/g [52].

#### 3.2.3. Effect of Adsorbent Amount on Cu^2+^ Adsorption of MGO@ZIF-8

The influence of the dosage of adsorbent on adsorption is one of the important indices to evaluate the properties of adsorbent materials. Consequently, it is of great significance to explore the optimal feeding ratio of materials to maximize their utilization rate. Figure 10 shows the comparison of Cu^2+^ adsorption effects of ZIF-8, MGO and MGO@ZIF-8 at different dosages. Obviously, among the three samples, MGO showed the worst adsorption effect on Cu^2+^, while the adsorption effect of ZIF-8 was better than the composite MGO@ZIF-8 when the amount of material was 1.0–2.0 mg, which may have been due to the better dispersion of the appropriate amount of ZIF-8 in the adsorption system [53]. However, the adsorption of Cu^2+^ by MGO@ZIF-8 was superior to pure ZIF-8 when MGO@ZIF-8 and ZIF-8 were at the same amount, which was due to the synergistic effect between MGO and ZIF-8. In addition, ZIF-8 did not show agglomeration under the influence of MGO, providing more adsorption sites.

#### 3.2.4. Time-Dependent Adsorption Behavior

The adsorption kinetic of MGO@ZIF-8 to adsorb Cu^2+^ at about 25 °C was also investigated. As shown in Figure 11, the adsorption of the MGO@ZIF-8 for Cu^2+^ showed different time-dependent behaviors, and the equilibrium for Cu^2+^ was attained within 2 h. Initially, it was clear that the growth rate of the adsorption capacity of MGO@ZIF-8 at 0–70 min increased significantly fast. After 70 min, the adsorption rates of MGO@ZIF-8 decreased to a certain extent, approaching the adsorption equilibrium, verifying that the combination of MGO and ZIF-8 could improve the adsorption effect of ZIF-8 [54].

#### 3.2.5. Analysis of Orthogonal Experiment Results

In order to explore the optimum condition for MGO@ZIF-8 to adsorb copper ions, the pH, adsorption temperature, quantity of adsorbent and adsorption time were selected to design research, based on the four factors and three levels of an orthogonal experiment. The selection of horizontal conditions for each factor are displayed in Table 1, and the experimental results were intuitively calculated and analyzed, as shown in Table 2. The adsorption amounts were analyzed, and the results are shown in Table 2. The adsorption of copper ions by MGO@ZIF-8 was optimal when the pH value was 6, the adsorption time was 70 min, the adsorption temperature was 35 °C, and the adsorbent was 5 mg. As displayed in Table 3, the Cu^2+^ adsorption capacity with the reported materials were compared, and the results showed that the Cu^2+^ adsorption capacity of MGO@ZIF-8 prepared in this work was higher than the materials described in previous literature.

#### 3.2.6. Adsorption Kinetics

The quasi-first-order and quasi-second-order fitting results of the time curve for Cu^2+^ adsorption effect of MGO@ZIF-8 are shown in Figure 12, and copper ion adsorption kinetic parameters of MGO@ZIF-8 were obtained according to the fitting results. As shown in Table 4, the actual maximum adsorption equilibrium quantity was 431.63 mg/g, and the theoretical adsorption quantity was 489.82 mg/g, based on the quasi-first-order kinetic fitting, which was quite different from the actual result. However, during the quasi-second-order kinetic fitting of the adsorption time curve, the theoretical maximum adsorption capacity was 432.50 mg, which was consistent with the experimental result, and the quasi-second-order fitting coefficient R^2^ was 0.997 > 0.983, as shown in Figure 13 and Table 4, thus, the quasi-second-order fitting result was good, because the MGO@ZIF-8 had the expected Cu^2+^ adsorption capacity [56]. Based on the fitting results, the adsorption process of MGO@ZIF-8 was in better agreement with the quasi-second-order kinetics, and the adsorption rate was positively correlated with the quadratic power of the maximum adsorption capacity under appropriate adsorption conditions. These results indicated that the Cu^2+^ adsorption behavior of the MGO@ZIF-8 adsorbent well-fitted the pseudo-second-order equation, in which the adsorption process was controlled by the chemical adsorption mechanism.

#### 3.2.7. Adsorption Isotherm

More importantly, in order to estimate the Cu^2+^ adsorption capacity of MGO@ZIF-8 on the basis of Cu^2+^ concentration and the parameters, the adsorption isotherm was performed. Figure 14 and Table 5 list the parameters and fitting condition for the Langmuir and Freundlich models, in which it can be seen that the Cu^2+^ adsorption capacity of MGO@ZIF-8 significantly increased before the initial Cu^2+^ concentration increased to 70 mg/L, then was close to the stable values, as shown in Figure 14a,c. Table 5 shows that although both models were suited to fit the Cu^2+^ adsorption isotherm by MGO@ZIF-8, the Langmuir model delivered a higher correlation coefficient of R^2^ (0.999), which confirmed that the adsorption process was controlled by chemical interactions and took place on the homogeneous surface of the MGO@ZIF-8. In addition, the theoretical maximum Cu^2+^ adsorption capacity of 458.2 mg/g was obtained according to the Langmuir model.

## 4. Conclusions

In conclusion, ZIF-8 was loaded into the interlayer of MGO by a simple in situ growth, to prepare MGO@ZIF-8. The maximum adsorption capacity of MGO@ZIF-8 achieved 431.63 mg/g when pH was 6, the temperature was 35 °C, and the adsorption time was 70 min. The excellent adsorption was attributed to the layered structure of graphene that provided ZIF-8 with uniformly distributed growth sites. The adsorption law of Cu^2+^ on MGO@ZIF-8 complied with the pseudo-second-order kinetic model and the Langmuir model, which confirmed that the process was monolayer chemisorption and a spontaneous endothermic process. The simple preparation process effect helped obtain MGO@ZIF-8 with high adsorption, which has potential application prospects in the field of heavy metal treatment.

## Figures and Tables

**Figure 1 nanomaterials-12-03162-f001:**
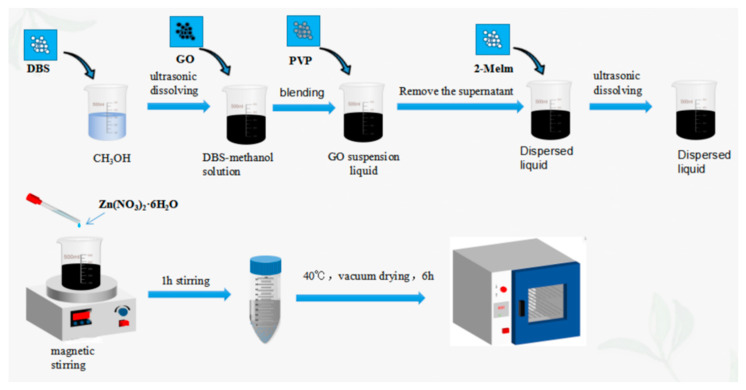
The preparation process of MGO@ZIF-8.

**Figure 2 nanomaterials-12-03162-f002:**
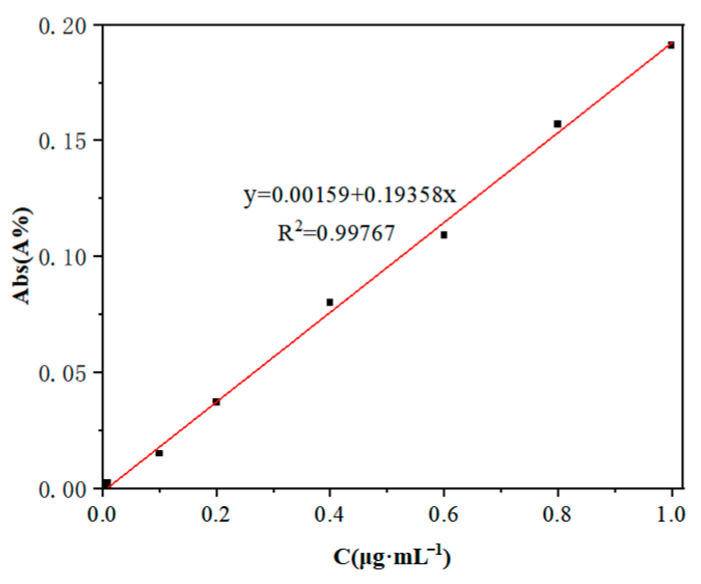
Standard curve of Cu^2+^.

**Figure 3 nanomaterials-12-03162-f003:**
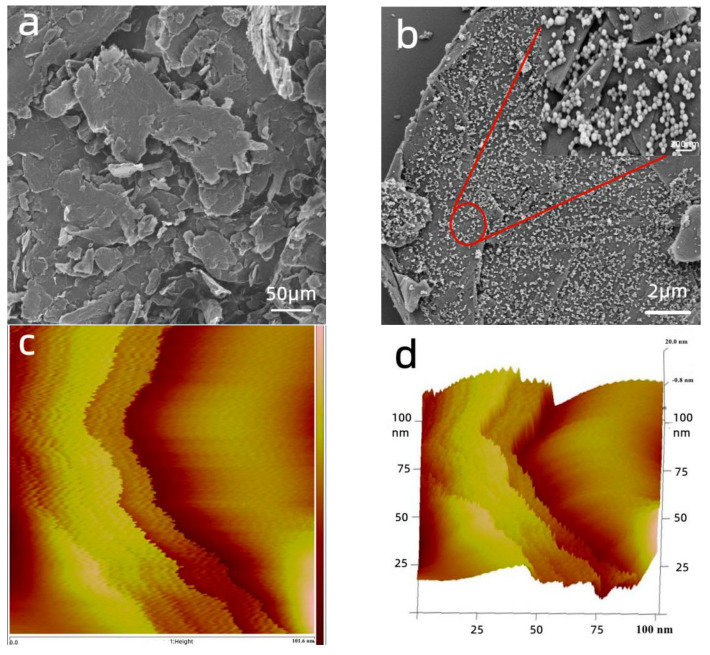
SEM images of the samples: (**a**) MGO, and (**b**) MGO@ZIF-8; and AFM images of the samples: (**c**) MGO@ZIF-8, and (**d**) MGO@ZIF-8.

**Figure 4 nanomaterials-12-03162-f004:**
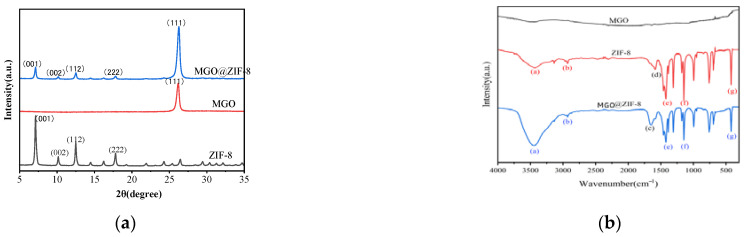
(**a**) XRD patterns of the samples; (**b**) FT-IR spectra analysis.

**Figure 5 nanomaterials-12-03162-f005:**
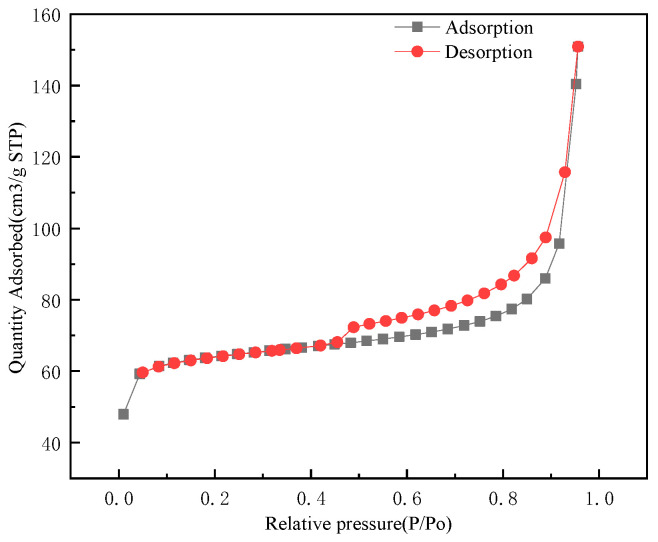
Isotherms of MGO@ZIF-8.

**Figure 6 nanomaterials-12-03162-f006:**
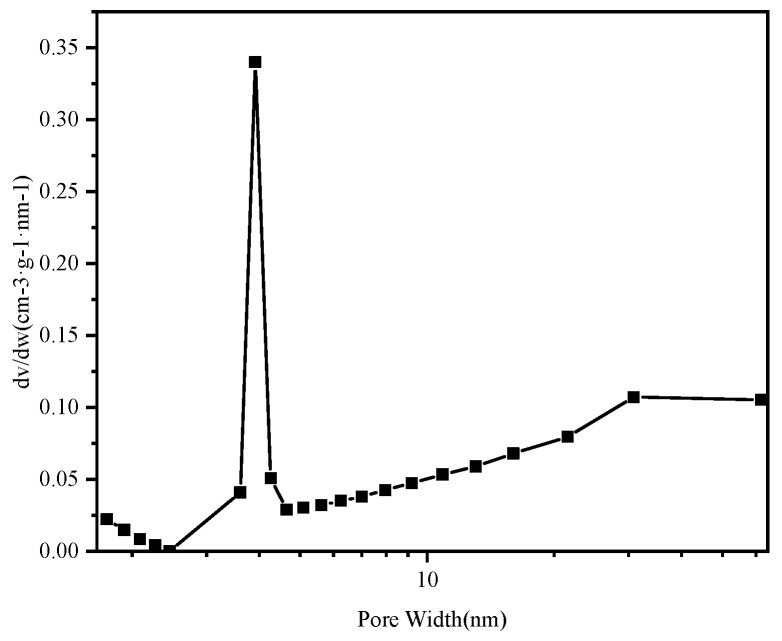
Aperture distribution curve of MGO@ZIF-8.

**Figure 7 nanomaterials-12-03162-f007:**
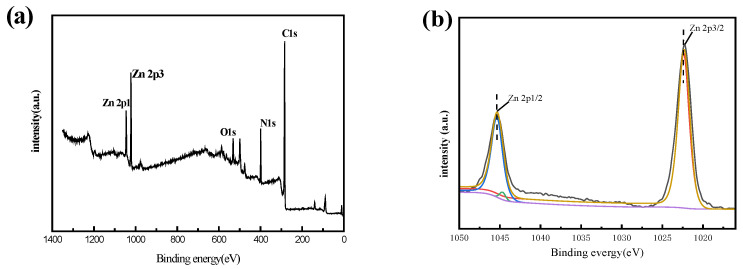
XPS analysis of: (**a**) survey scan of MGO@ZIF-8, (**b**) Zn 2p, (**c**) C 1s, (**d**) O1s,(**e**) survey scan of Cu-MGO@ZIF-8, and (**f**) Cu 2p.

**Figure 8 nanomaterials-12-03162-f008:**
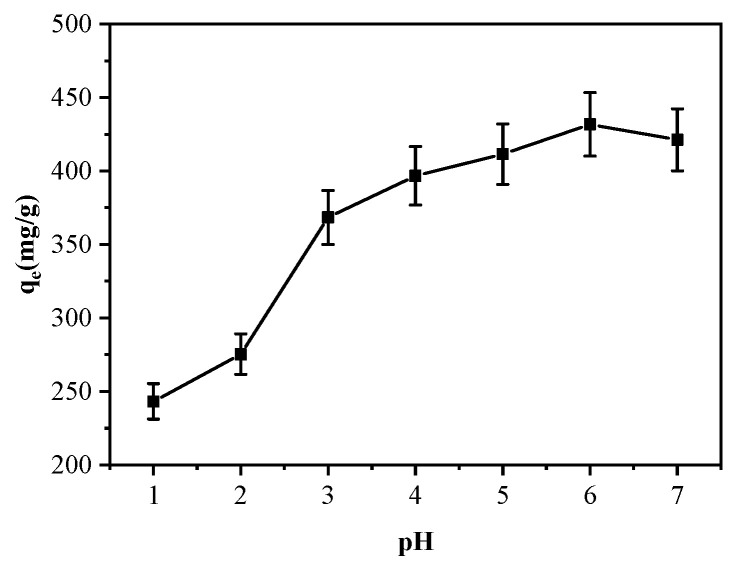
Effect of pH on Cu^2+^ adsorption.

**Figure 9 nanomaterials-12-03162-f009:**
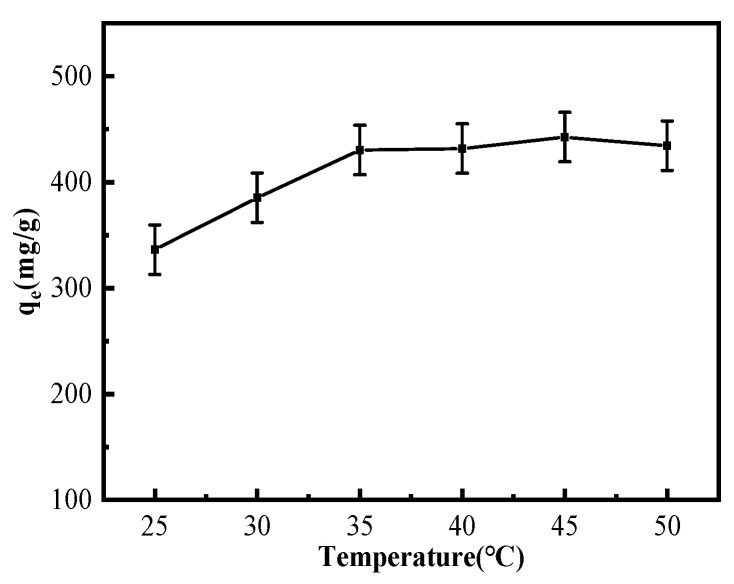
Effect of temperature adsorption.

**Figure 10 nanomaterials-12-03162-f010:**
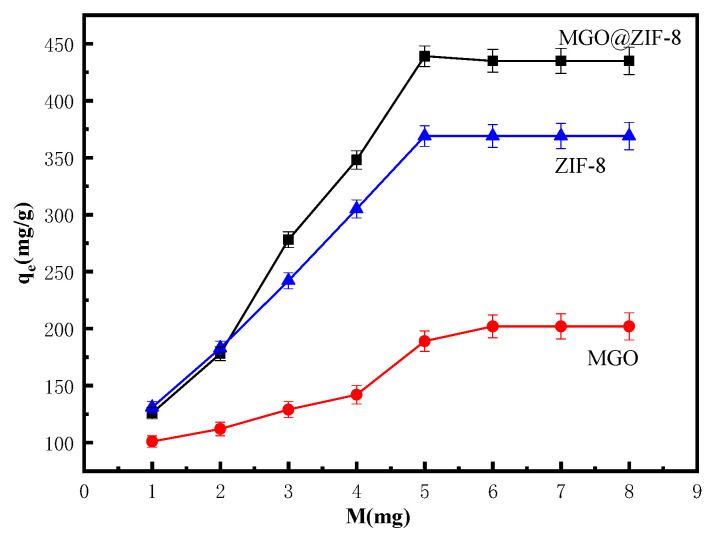
Effect of amount of adsorbent on adsorption.

**Figure 11 nanomaterials-12-03162-f011:**
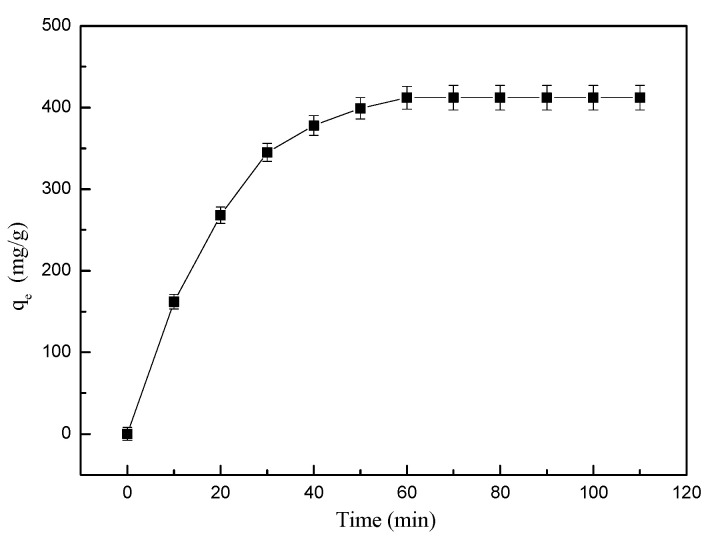
Time-dependent adsorption behavior of MGO@ZIF-8.

**Figure 12 nanomaterials-12-03162-f012:**
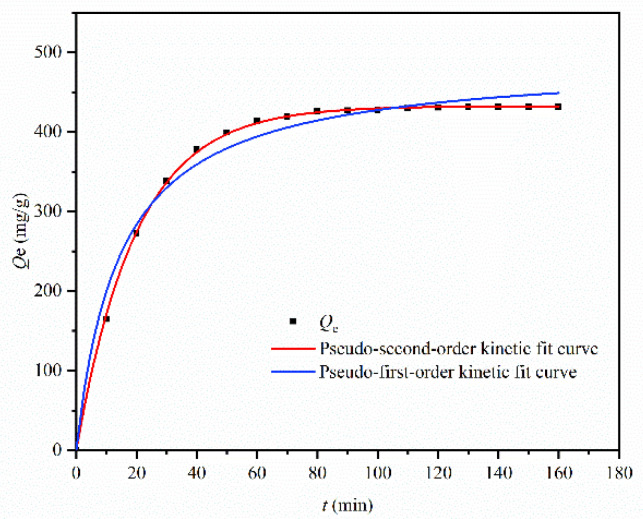
Pseudo-first-order and pseudo-second-order nonlinear fitting of dynamics.

**Figure 13 nanomaterials-12-03162-f013:**
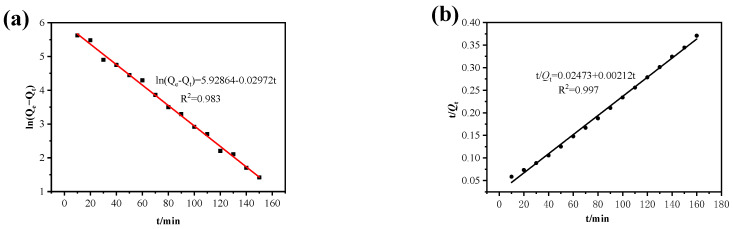
Pseudo-first-order (**a**), and pseudo-second-order (**b**), linear fitting of dynamics.

**Figure 14 nanomaterials-12-03162-f014:**
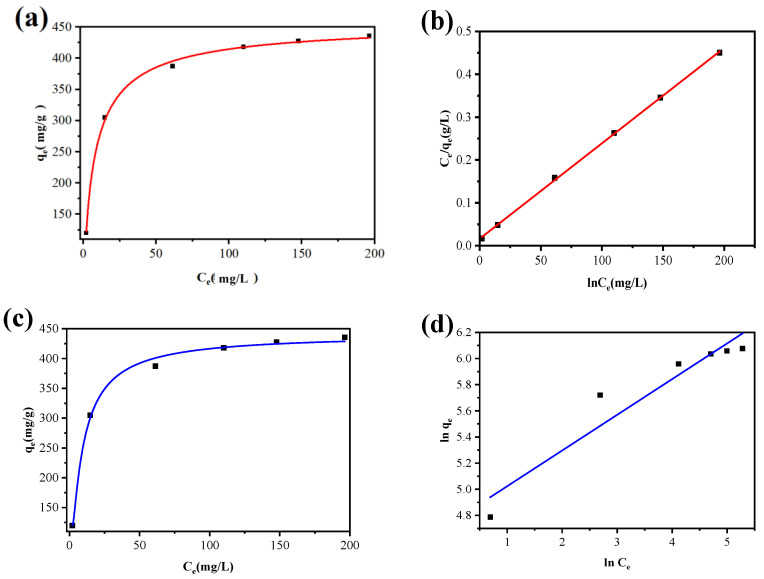
(**a**) Langmuir nonlinear fitting; (**b**) linear fitting for MGO@ZIF-8 adsorption of copper ions; (**c**) Freundlich nonlinear fitting; and (**d**) linear fitting for copper ion adsorption.

**Table 1 nanomaterials-12-03162-t001:** Selection of experimental factor level conditions.

Level	pH	Time (min)	Temperature (°C)	Amount of Adsorbent (mg)	Results (mg/g)
1	5	50	30	4	Q
2	6	60	35	5	
3	7	70	40	6	

**Table 2 nanomaterials-12-03162-t002:** The experimental results are intuitively calculated and analyzed.

Number	Influence Factor				Adsorbing Capacity (mg/g)
	pH	Time (min)	Temperature (°C)	Amount of Adsorbent (mg)	
1	5	50	30	4	390.05
2	5	70	35	5	437.21
3	5	70	40	6	410.37
4	6	50	35	4	420.45
5	6	60	40	5	400.52
6	6	70	30	6	440.83
7	7	50	40	5	403.16
8	7	60	30	6	384.91
9	7	70	35	2	397.42
*K* _1_	412.333	404.333	404.667	395.677	——
*K* _2_	420.000	407.000	418.000	426.667	——
*K* _3_	394.667	415.667	404.333	404.667	——
R	25.333	11.334	13.667	31.000	——

**Table 3 nanomaterials-12-03162-t003:** Comparison of Cu^2+^ adsorption capacity of MGO@ZIF-8 with similar and other previously used adsorbents.

Sorbent	Adsorption Capacity	pH	Temperature (°C)	Reference
CD-CA/PDA	73.46	6	25	[10]
Magnetite Nanoparticles	6.28	7	30	[12]
Fe_3_O_4_@zeolite NaA	86.54	4	25.15	[6]
Tyr-Mt	28.31	5	15	[11]
Modified latex sponge	125.8	5.85	40	[38]
PVA/GO	44.7	5.8	50	[47]
GO/PEI/CMC	302.04	5.0–5.5	25	[55]
MGO@ZIF-8	431.63	6	35	This work

**Table 4 nanomaterials-12-03162-t004:** Copper ion adsorption kinetic parameters of MGO@ZIF-8.

Copper Ion	*Q*_e_ (mg/g)	Pseudo-First-Order			Pseudo-Second-Order		
		*K*_1_ (min^−1^)	*Q*_e_ (mg/g)	*R* ^2^	*K*_2_ (g/(mg·min))	*Q*_e_ (mg/g)	*R* ^2^
Cu	431.63	0.05	489.82	0.983	0.0014	432.50	0.997

**Table 5 nanomaterials-12-03162-t005:** Adsorption isothermal model parameters of MGO@ZIF-8 for Cu^2+^.

——	Langmuir			Freundlich		
*T*/K	*Q* _e_	*K* _L_	*R* ^2^	*K_F_*	1/n	*R* ^2^
308	458.20	0.016	0.999	0.27318	0.653	0.963

## Data Availability

Data is contained within the article.

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
