# Peer review of "Multilayer Graphene Oxide Supported ZIF-8 for Efficient Removal of Copper Ions"

_nanomaterials, 2022, doi:10.3390/nano12183162_

Round 1

Reviewer 1 Report

The work "Multilayer graphene as a support to synthesize ZIF-8 adsrobents for Cu2+ removal" needs some major corrections at this stage before consideration of publication in this journal.

1.    Why didn’t try Fig 1 is not clear.

2. Page 7, row 171: 421 cm-1 due to Zn-N bending vibrations and not C-H stretching.

3.     Page 9, row 203: please correct the values of C1s spectrum. Is it 284.18 eV and not 384.18 eV?

4.     What about C=C peaks in C1s spectrum

5.     The mechanism of storage mechanism is not accurate and detail enough.

6.     There are some grammar errors and inappropriate expressions.

7.     The present results should compare with another researches or simulations in the form of table.

8.     Cu2+ is removal is not big advantage compare to heavy toxic metals from industrial water(Pb, As, Hg, Cd, Cr).

9.     Finally, the novelty of the paper is relatively unclear. And the content of the study is very basic and  shallow. further deeper studies are required to improve the novelty of the study.

Reviewer 2 Report

Reviewer’s comment:

This work by Lv et.al showed the utilization of ZIF-8 and multi-layer graphene nanocomposites for removing Cu2+ ions and has taken into consideration of various factors that affect the process. The study is detailed and has obtained good results. However, there are some problems and issues which are mentioned in the comments below that need to be addressed before they can be considered for publication in nanomaterials.

1.       The preparation of MGO@ZIF-8 should be completely described. Such as the amount of 2-Melm and Zinc precursors used during the process. The amount of solvent, ligand, and metal source plays a critical role in the shape and size of the MOF. So, the authors should describe the critical parameters properly.

2.       What is the optimum loading of ZIF-8 on MGO for the best Cu2+ absorption? Did the authors prepare MGO@ZIF-8 at different concentrations of Zn and 2-Melm and check its effect on the absorption capacity of Cu2+?

3.       What is the solvent used for making the Cu2+ solution? Is it methanol or water? It should be mentioned.

4.       What chemical is used for changing the pH of the Cu2+ solution? It should be mentioned under the experimental section in 2.1 materials sub-section.

5.       The description for the assessment process of MGO@ZIF-8 for Cu2+ adsorbing capacity in section 2.3 is not satisfactory. More details of the process should be explained. Such as the authors mention seven conical bottles without giving any details in the previous sentences.

6.       The values of the relative pressure (P/Po) in Figure 5 seem totally wrong. How can the values of (P/P0) be as high as 800?

7.       Although the authors claimed that the nitrogen isotherm of MGO@ZIF-8 belonged to type II isotherm. However, considering the hysteresis loop and the pore diameter being mesopore suggest that typical type 4a with hysteresis loop H4 is the more correct type of the isotherm curve. The authors can refer the analysis from Chem. Soc. Rev. 2017, 46, 389.

8.       The specific surface area (SSA) and the porosity of MGO and ZIF-8 should also be investigated for comparison with those of MGO@ZIF-8.

9.       The image of the magnetic stirring part in the scheme should be checked properly. Some Chinese letters are visible.

 10.   The change in the morphology and the phase of the MGO@ZIF-8 after the absorption test should also be investigated through XRD and FE-SEM.

Round 2

Reviewer 1 Report

Accept

Reviewer 2 Report

The authors have revised the manuscript properly by answering to all the comments. So, I recommend to accept the revised manuscript for publication in Nanomaterials.